# Characterization and Analysis of Chitosan-Gelatin Composite-Based Biomaterial Effectivity as Local Hemostatic Agent: A Systematic Review

**DOI:** 10.3390/polym15030575

**Published:** 2023-01-22

**Authors:** Heri Herliana, Harmas Yazid Yusuf, Avi Laviana, Ganesha Wandawa, Arief Cahyanto

**Affiliations:** 1Doctoral Program, Faculty of Dentistry, Universitas Padjadjaran, Bandung 45124, Indonesia; 2Department of Oral and Maxillo Facial Surgery, Faculty of Dentistry, Universitas Padjadjaran, Bandung 45124, Indonesia; 3Department of Orthodontic, Faculty of Dentistry, Universitas Padjadjaran, Bandung 45124, Indonesia; 4The Indonesian Naval Dental Institute, Jakarta 10210, Indonesia; 5Department of Dental Materials Science and Technology, Faculty of Dentistry, Universitas Padjadjaran, Bandung 45124, Indonesia; 6Department of Restorative Dentistry, Faculty of Dentistry, University of Malaya, Kuala Lumpur 50603, Malaysia

**Keywords:** chitosan-gelatin, composite material, hemostatic agent

## Abstract

Chitosan and gelatin were the most widely used natural materials in pharmaceutical and medical fields, especially as local hemostatic agents, independently or as a composite material with the addition of other active substances. Chitosan and gelatin have excellent properties in biocompatibility, biodegradability, non-toxicity and water absorption capacity. The objective of this review was to analyze the characteristics of chitosan-gelatin (CG) composite-based biomaterial and its effectivity as a local hemostatic agent. We used PRISMA guidelines and the PICO framework to compile this review. The findings demonstrated that the CG composite-based biomaterial had excellent physical, chemical, mechanical properties and local hemostatic agent activity by adding other active substances such as oxidized fibers (OF), silica nanoparticles (SiNPs), calcium (Ca) and biphasic calcium phosphate (BCP) or by setting the CG composite proportion ratio.

## 1. Introduction

Uncontrolled bleeding is a medical emergency that can be life-threatening. This condition can occur due to soft and hard tissue trauma, disrupting the continuity of blood vessels. Trauma that causes bleeding can occur in everyday life, such as in traffic accidents, sports, work accidents, fights, complications of surgery and specifically on wounds that occur on the battlefield. The percentage of initial causes of death in trauma cases in the life of civil society is around 80% due to uncontrolled bleeding. Meanwhile, in the last 150 years, the death rate of soldiers on the battlefield has been around 20%, of which half of the causes of death are due to exsanguination [1,2,3]. The development of new methods, tools and materials for bleeding control can significantly reduce morbidity and mortality in cases of uncontrolled bleeding in the future [4].

Research and development of hemostatic methods and drugs, especially local hemostatic, is currently being carried out either by finding new materials or making modifications to improve existing materials. One of the local hemostatic agents that are widely used today is a mechanical hemostatic type of gelatin material in the form of an absorbent sponge. Gelatin is one of the essential natural biopolymers resulting from the denaturation of collagen-derived proteins through a partial thermohydrolysis process and has the property to change shape reversibly between sol and gel [5]. Sources of gelatin are still dominated by gelatin derived from mammalian animal tissues, generally pigs and cows [6,7]. The world’s sources of gelatin are 46% pork skins, 29.4% cow skins, 23.1% beef bones and 1.5% other sources such as poultry and fish [8]. Using gelatin from mammals such as pigs and cows has limitations according to religious and infectious disease issues. It is prohibited for both Moslem and Jewish communities to consume all pork-based products, and Hindu communities would reject bovine-based products. Furthermore, bovine spongiform encephalopathy (BSE) in cows and swine flu in pigs raise safety concerns [9,10,11].

Another biomaterial that has a local hemostatic effect is chitosan, a natural polysaccharide of animal origin synthesized from chitin by partial deacetylation. The main sources of chitin are found in animal shells such as crustaceans, insect cuticles and yeast [12,13]. Gelatin and chitosan are two types of biomaterials that are widely used in the medical, pharmaceutical, cosmetic and food processing fields because these two biomaterials have advantages in terms of being biocompatible, biodegradable, non-toxic, fewer allergic reactions and having a hemostatic effect. In addition, chitosan also has an antibacterial effect [14,15,16]. 

The combination of two materials with different physical, chemical and mechanical properties to form a new material with superior properties and characteristics is called a composite material [17]. The application of chitosan-gelatin (CG) composite-based biomaterials has been widely used as a local hemostatic agent and in other biomedical fields and food packaging processes [18,19,20,21,22]. For example, Ali et al. reported a study about antimicrobial and wound-healing activities of graphene-reinforced electrospun chitosan/gelatin nanofibrous nanocomposite scaffolds [23]. In the food packaging process, Wang et al. reported a study about edible films from CG: physical properties and food packaging applications [24].

CG nanocomposites are rapidly emerging among the wide range of materials available for various biomedical applications. This nanocomposite is typically created by incorporating nanoparticles into a CG matrix via various crosslinking methods. Aside from acting as a matrix for dispersing nanoparticles, the CG composite can also act as a reducing and stabilizing agent for nanoparticles to prevent agglomeration. Chitosan and gelatin are frequently mixed to enhance the biological features of the material produced by stimulating cell attachment and the development of a polyelectrolyte complex [21]. 

This systematic review aims to analyze the properties of CG composite-based biomaterial and its potency as an excellent local hemostatic agent, then use the results as a reference to find alternative sources of gelatin to replace conventional gelatin from mammals such as pigs and cows that have limitations due to religious and infectious disease issues.

Many studies have been conducted to analyze the effectiveness of CG composite-based biomaterials as local hemostatic agents. These studies showed that the CG composite has a local hemostatic effect that can be increased by adjusting the composition of the material percentage, adding other substances or materials and using different combining methods. These many studies became the background for the researcher to conduct a systematic literature review with the research question: “How are the characteristics and effectiveness of the CG composite-based biomaterial as a local hemostatic agent”?

## 2. Methods

### 2.1. Search Strategy and Paper Selection

This systematic review was conducted using a research protocol that refers to PRISMA (Preferred Reporting Items for Systematic Reviews and Meta-Analyses) guidelines as presented in Figure 1 [25]. Inclusion criteria from the literature reviewed were compiled based on the PICO (Problem/Population, Intervention, Comparison, and Objective) framework [26]. Problems include bleeding cases, intervention using CG composite-based biomaterial, and comparison using a control group of other materials or commercial materials. The objective is to assess the effectiveness of CG composite-based biomaterial as a local hemostatic agent based on the characteristics of the material and analysis of its effectiveness in the hemostasis process. The literature is research articles, both laboratory tests and clinical trials, published in the last ten years in reputable international journals, and the language used is English. The literature search method uses a combination of keywords based on Boolean operators. The keywords used in the search were: Composite AND “Chitosan-Gelatin” AND Biomaterial AND (Hemostatic OR Hemostasis).

The search for the literature was carried out from January to May 2022 on international journal provider database sites, including Scopus, PubMed, and ScienceDirect. We found 243 articles containing studies about CG composite-based biomaterial related to the hemostasis process. The screening process was conducted manually and partly assisted by the Mendeley application. Two reviewers (HH and AC) examined the complete lists of results from the three database searches for eligibility, and disagreements were settled through discussion and consensus. Following the systematic screening that resulted in four eligible articles for review. The stages of the search process to obtain eligible articles are presented in Figure 1 using a PRISMA flow diagram. This systematic review has been registered at PROSPERO with the register number CRD42022362010.

### 2.2. Assessment of Methodological Quality and Risk of Bias 

The quality assessment was carried out independently by two authors, HH and GW, using SYRCLE’s risk of bias tool, as shown in Table 1 [27]. A ‘Yes’ (Y) indicates a low risk of bias in a specific domain, whereas a ‘No’ (N) indicates a high risk of bias in that domain. We used ‘Unclear’ (U) when there was insufficient information to assess the risk of a particular bias.

## 3. Results

From 243 articles found in databases, only four articles meet the inclusion criteria. All research articles were Q1 Scopus indexed and published in the last ten years, with details of one article published in 2015, one published in 2019 and two published in 2021. The research methods used were all in vitro and in vivo experimental studies. The following are the characteristics of the four included studies as shown in Table 2.

The following Figure 2 shows the schematic view of CG-based composite material studies combined with OF, SiNPs, Ca, and BCP. The next step was in vitro characterization test using SEM and FTIR, and the final step was the in vivo study to analyze the hemostatic ability of the materials.

Data collection from each article was conducted by two reviewers (HH and AC) if there were any disagreements settled with discussion and consensus. Data analysis of the characteristics of physical, chemical and mechanical properties, as well as the biocompatibility and hemostatic test of CG composite-based biomaterial from four studies, are presented in Table 3.

Table 3 above shows that three articles describe the study of CG composite with the addition of other active substances and evaluate its characteristics and effectiveness as a hemostatic agent. Jalal Ranjbar et al. reported that the biocomposite sponge added by 30% Oxidized Fibers (OF) suspension to CG solution had the least amount of bleeding and the best bio-absorbability after 21 days. The addition of OF to CG composite boosted the clotting ability and bio-absorbability of the biocomposite sponges [28]. It proved the previous study by Zhang et al. reported that oxidized cellulose-based hemostatic material’s main advantage is bio-absorbability and biodegradation in the human organism, which can be tailored depending on the structure of the original cellulose and degree of oxidation [32]. 

Gokul Patil et al. conducted a study on the fabrication of CG composite xerogel incorporated with silica nanoparticles (SiNPs) and calcium (Ca). The results stated that the composite has good platelet activation and thrombin generation. It had a high compressive strength of 2.45 MPa and could bear pressure while in use. Furthermore, the xerogel composite’s biocompatibility was excellent. In vivo application of xerogel composite to a deadly femoral artery lesion in rats resulted in hemostasis (2.5 min), which was significantly faster than commercial CELOX (CX) (3.3 min) and Gauze (4.6 min), and the wound was easily removed. After being irradiated, the gamma-irradiated composite remained stable for 1.5 years [29].

Another article studied CG combined with other substances was by Padalhin AR and Lee BT, which reported that the bi-layered topical hemostat composed of electrospun gelatin loaded with biphasic calcium phosphate (BCP) and chitosan could stop bleeding within 3 min of application to the bleeding location and dramatically improve bone regeneration. Using the bi-layer material as a degradable hemostat also significantly increased bone healing in the irradiated 3 mm defect [30].

Meanwhile, only one article described the study of the characterization of CG composite biomaterial and its hemostatic activity without adding any other substances, as Guangqian Lan et al. reported. This study fabricated a composite sponge containing chitosan and gelatin in different proportions. The optimum blood clotting index (BCI) was reached in vitro by a chitosan/gelatin sponge (CG) ratio of 5/5 (*w*/*w*), according to the findings. Furthermore, compared to the two components independently, CG showed the best hemostatic efficacy in rabbit artery bleeding and liver model tests [31].

The risk of bias analysis using SYRCLE’s tool revealed that the overall article reviewed mostly has an unclear bias, and authors stated that their institution’s authority had approved each protocol, as shown in Figure 3 below. Except for baseline characteristics, all studies have a low risk of bias. Sequence generation and random housing revealed that 50% or two studies have a low risk of bias and one study has a low risk of bias in random outcome assessment. 

## 4. Discussion

The most common cause of death in trauma patients is bleeding. Hemorrhages caused by military conflicts, vehicle accidents, and trauma causes many troops and people to die yearly. Controlling bleeding correctly and quickly can save time for follow-up therapy and, as a result, minimize mortality. Several types of quick hemostasis methods exist in this context, and rapid hemostatic materials have become essential for clinical and emergency therapies [33,34]. 

Chitosan and gelatin were biomaterials from natural sources that were most widely used in pharmaceutical and medical fields, especially for hemostatic purposes, because they had similar good properties that matched the criteria needed for local hemostatic agents, such as biocompatibility, biodegradability, non-toxic and less allergic effects [35]. Many studies showed that chitosan triggered coagulation without activating the intrinsic pathway, indicating that the hemostatic mechanism of chitosan was independent of the classical coagulation cascade [36]. Meanwhile, gelatin-based sponge matrix possesses a unique microstructure with high porosity, high surface roughness in the nano range, faster blood absorption, and rapid hemostasis [37,38].

The most crucial characteristic of a hemostatic sponge during the hemostasis process is water or blood absorption capacity. To achieve hemostasis during heavy blood loss following an injury, any applied hemostatic material must absorb blood quickly and induce the rapid formation of a blood clot. Chitosan and gelatin are both known hemostatic agents that work by stimulating platelet aggregation. However, both have limitations in terms of reducing hemostatic effects. Chitosan is poorly absorbed and soluble, whereas gelatin has a low absorption potential. Several studies reported that mixing chitosan and gelatin could improve the limitations of chitosan and gelatin if used as a single material [28,29,30,31,32,33]. Furthermore, the association of chitosan and gelatin, when loaded by nanoparticles, for instance, bioactive nanoparticle material, could form nanocomposites that enhance biocompatibility, hemostatic potential, and bioactivity, that are suitable candidates for biomedical application, including hemostatic agents [21]. 

This systematic literature review analyzed four articles about the characterization and evaluation of CG composite-based biomaterial’s hemostatic or blood coagulation activity. We found only four articles that met the inclusion criteria from three international journal databases. This number showed that the study about the function of CG composite-based biomaterial as a hemostatic agent is still limited. Chitosan and gelatin could be used as a local hemostatic agent either independently or blended with other active substances to increase the hemostatic activity of the material, as seen in Table 3 above. 

The characterization of CG composite from four articles showed different results, but in general, all studies concluded that CG composite has better physical, chemical and mechanical properties than the control group. At the same time, the hemostatic ability of CG composite could be improved by adding other active substances. The addition of oxidized cellulose fibers to the CG composite represents a new class of biodegradable biopolymers that could be used as hemostatic materials to control low to moderate bleeding. They instantly drain fluid from the circulation and concentrate coagulation components, resulting in a faster coagulation process [28]. 

Incorporating hemostatic agents such as SiNPs and Ca work synergistically with CG xerogel to exert stronger hemostasis in a rat lethal femoral artery damage model, demonstrating that the xerogel composite has better hemostatic properties than commercially available CX and Gauze [29]. Due to their exothermic reaction, hemostatic dressings such as CX and QuickClot can potentially burn the wound site [39,40]. However, the newly developed xerogel composite may be easily withdrawn from the wound site without breaking the clot or causing repeated bleeding.

A bi-layer composite hemostatic material containing CG composite incorporated with BCP was designed, fabricated and tested specifically for hemostatic and bone tissue application. Filtering and aggregating the majority of the cell components along with the electrospun layer of the composite chitosan/gelatin-BCP material would then aid in forming a stable plug to prevent further bleeding on the bone defect [30]. The chitosan hydrogel layer behaves like a sponge and may absorb more fluid than the fibrous gelatin layer because of its three-dimensional porosity [41].

CG composite, without the addition of any other substances, could be improved its physical, chemical, and mechanical properties as well as its hemostatic effectiveness by setting the proportion of both materials. When compared to chitosan or gelatin materials alone, the composite had a steadier three-dimensional structure and more even perforations, making it superior in boosting platelet aggregation and holding erythrocytes to endorse blood clotting and create a stable clot in amorphous material to produce quick hemostasis. The proportion of 5/5 (*w*/*w*) CG ratio showed the best result in the blood clotting index test [31].

In this study, CG composite-based biomaterial is considered one of the essential substances since CG has been widely used as raw material for some products in the medical, pharmaceutical and food packaging industries in the last ten years [42]. Especially gelatin is a fibrous protein derived from mammalian, avian, and fish species collagen through partial hydrolysis. Today, raw materials used in gelatin production are cattle bones, hides, and pork skin. In addition to fish skin, fish bone and chicken bone can be used as raw materials in gelatin production [43,44,45].

Gelatin has a variety of biomedical applications, including tissue engineering. It is preferred over collagen for biomedical applications because collagen is known to be antigenic; gelatin is not antigenic and is one of the most convenient proteins for such applications [46,47,48]. Another essential function of gelatin in biomedical is used as a local hemostatic agent. Hemostatic agents are frequently used in surgery because they assist a patient’s coagulation system in rapidly developing an occlusive clot. These agents are used for intraoperative hemostasis, hemorrhage control in dental extractions and oral surgery, intestinal, orthopedic, and ear surgery, as well as microcoil embolization in upper and lower gastrointestinal hemorrhage [43,49]. Although the effectivity of CG composite-based biomaterial as a local hemostatic agent has been reported to have significant results in all studies, the mechanisms of action in the hemostasis process remain unclear. 

## 5. The Advantages and Mechanism of Chitin/Chitosan and Gelatin for a Hemostatic Agent

The primary source of chitosan synthesis is chitin deacetylation. At both room and elevated temperatures, strong alkali solutions are used to remove N-acetyl groups during the deacetylation process. It is a natural, non-immunogenic, biodegradable, non-toxic and mucoadhesive polysaccharide that has been studied for various biomedical applications [50,51]. Chitosan has shown promise as a wound dressing material due to its hemostatic and antibacterial properties. The hemostasis activity is determined by a dynamic balance of anticoagulant and coagulating substances in the blood and blood vessels. A chitin-based hemostatic agent was recently used for the critical treatment of an open-wound hemorrhage. Chitosan has been shown to reduce in vitro blood clotting time by 40% compared to blood alone [52]. Chitosan is cationic due to positively charged glucosamine groups that can interact with the negative charge of the bacterial cell surface. The bacterial cell wall is ruptured as a result of this interaction. This can confer antibacterial activity on chitosan [53]. 

Gelatin is a denatured form of collagen that has been shown to have excellent properties as a material for various biomedical applications, including hemorrhage control. Gelatin’s hemostatic effect is attributed to swelling of the biopolymer upon contact with blood, as well as platelet activation and aggregation, which speeds up blood coagulation [49]. Gelatin, on the other hand, has poor mechanical properties at physiological temperatures. As a result, it is frequently combined with other materials, such as levodopa (L-DOPA), oxidized cellulose fibers, nanosilicates such as Laponite or polysaccharides such as chitosan [28,29,31]. The nanofiber sponge, made of gelatin aggregates, activated many platelets, promoting platelet embolism and escalating coagulation pathways, according to the researchers. Furthermore, in vivo studies demonstrated that these gelatin sponges could generate stable blood clots quickly with minimal blood loss [51].

Biopolymers have poor mechanical properties, chemical resistance, and processability compared to synthetic polymers. They are reinforced with fillers that significantly improve their inherent properties to make them suitable for specific applications. Biopolymer composites are biopolymer composites that have been reinforced in this way. They are combined with specific materials to strengthen and improve their desired properties for practical applications [54]. Chitosan has a low surface area and negligible porosity in its pure form and has disadvantages, including low mechanical strength and insolubility in water. To address these shortcomings, the molecule should be chemically and physically modified. The most common methods are chitosan crosslinking and grafting. Crosslinking is the process of creating a web between polymer strands to form a network. Grafting is the process of covalently bonding monomer chains to a polymer backbone. Crosslinked with gelatin is one of the most used methods to improve the properties of chitosan [55].

Recently, there have been many commercial products of chitosan-based and gelatin-based hemostatic agents in the market. For example, chitosan-based hemostatic materials such as HemCon^®^, Celox^TM^, and TraumaStat©. Gelatin-based hemostatic materials included Gelfoam^®^, Surgifoam/Spongostan^®^, Surgiflo^®^, and Floseal^®^ [56]. Unfortunately, there is no hemostatic agent based on CG composite found in the market or commercially produced. Therefore, it is important to give more attention to CG composite-based biomaterials as a hemostatic agent due to their excellent properties if blended.

## 6. Conclusions

All studies in this systematic review reported the superiority of CG composite-based biomaterials by characterization, biocompatibility and hemostatic activity tests. The addition of some other active substances, such as OF, SiNPs, Ca and BCP, could increase the superior properties of the CG composite as a local hemostatic agent. The other method to improve CG composite properties was by setting the material proportion ratio. Fish gelatin could be the most potent alternative to mammalian gelatin regarding the religious issues and infectious diseases generated by bovines and swine.

## 7. Perspectives and Future Direction

The natural source of chitosan in the studies above comes from marine crustaceans and silkworm pupae, while gelatin comes from bovine and porcine. Regarding religious issues (Moslem and Judaism forbid the consumption of pork-related products, and Hindus forbid the consumption of bovine products), as well as the potency of spreading diseases from those mammals such as bovine spongiform encephalopathy and swine flu, it is an opportunity to find another natural source of gelatin.

The byproducts of the fish processing industry can potentially be used in the production of gelatin. Gelatin from fish and other marine biotas could be the most potential alternative, but it needs more studies to find the best method to create a good quality fish gelatin similar to mammalian gelatin.

## Figures and Tables

**Figure 1 polymers-15-00575-f001:**
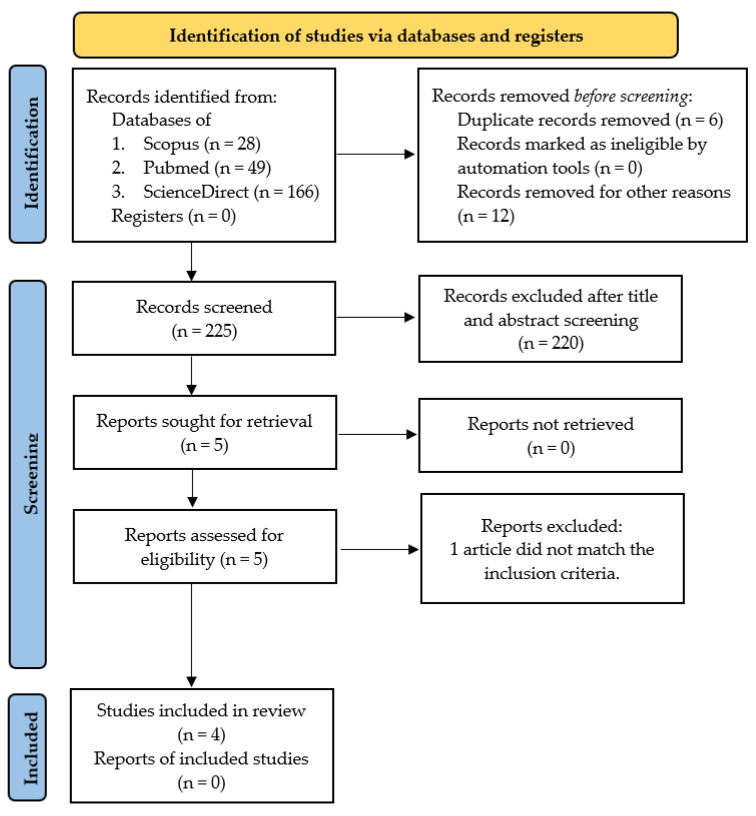
PRISMA flow diagram [25].

**Figure 2 polymers-15-00575-f002:**
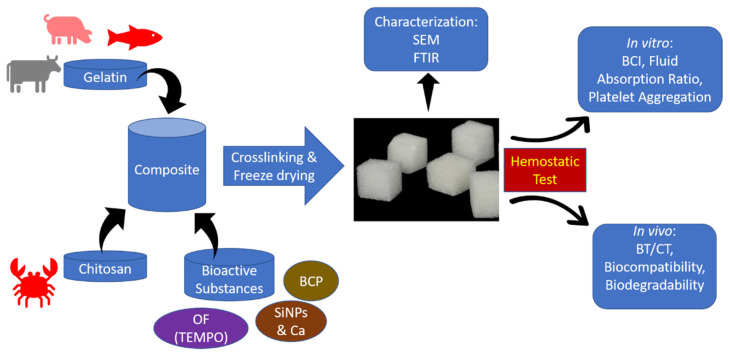
Schematic view of the studies (OF: Oxidized Fibers, SiNPs: Silica Nanoparticles, Ca: Calcium, BCP: Biphasic Calcium Phosphate, SEM: Scanning Electron Microscope, FTIR: Fourier Transform Infrared, BCI: Blood Clotting Index, BT/CT: Bleeding Time/Clotting Time).

**Figure 3 polymers-15-00575-f003:**
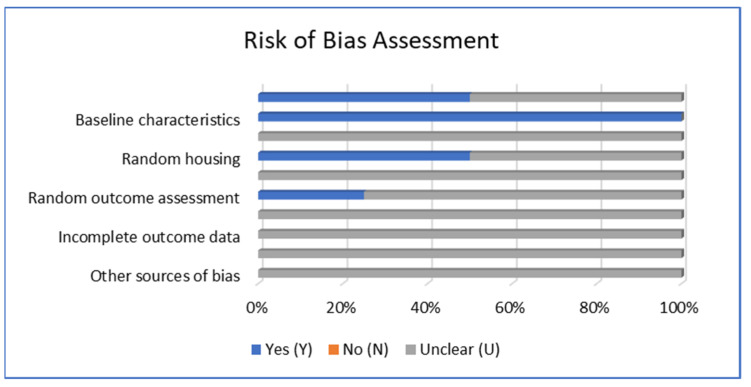
Risk of bias assessment using SYRCLE’s tool [27].

**Table 1 polymers-15-00575-t001:** SYRCLE’s Risk of Bias Tool [27].

Item	Type of Bias	Domain	Review Authors Judgement	Answer for Each Study			
				1 [28]	2 [29]	3 [30]	4 [31]
1	Selection bias	Sequence generation	Was the allocation sequence adequately generated and applied?	Y/Y	Y/Y	U/U	U/U
2	Selection bias	Baseline characteristics	Were the groups similar at baseline or was adjusted for confounders in the analysis?	Y/Y	Y/Y	Y/Y	Y/Y
3	Selection bias	Allocation concealment	Was the allocation adequately concealed?	U/U	U/U	U/U	U/U
4	Performance bias	Random housing	Are the animals randomly housed during the experiment?	U/U	Y/Y	U/U	Y/Y
5	Performance bias	Blinding Operation	Were the caregivers/and or investigators during the course of the experiment blinded from knowledge of which intervention each animal received?	U/U	U/U	U/U	U/U
6	Detection bias	Random outcome assessment	Were animals selected at random for the outcome assessment?	U/U	Y/Y	U/U	U/U
7	Detection bias	Blinding outcome assessment	Was the outcome assessor blinded?	U/U	U/U	U/U	U/U
8	Attrition bias	Incomplete outcome data	Were incomplete outcome data adequately addressed?	U/U	U/U	U/U	U/U
9	Reporting bias	Selective outcome reporting	Are reports of the study free of selective outcome reporting?	U/U	U/U	U/U	U/U
10	Others	Other sources of bias	Was the study apparently free of other problems that could pose a high risk of bias?	U/U	U/U	U/U	U/U

**Table 2 polymers-15-00575-t002:** Characteristics of Studies.

Authors, Years	Title	Journal/Index	Methods
Jalal Ranjbar et al., 2021 [28]	Novel chitosan/gelatin/oxidized cellulose sponges as absorbable hemostatic agents	Cellulose, 2021;28(6):3663–75. Q1	In vitro and in vivo experimental study
Gokul Patil et al., 2021 [29]	Design and synthesis of a new topical agent for halting blood loss rapidly: A multimodal chitosan-gelatin xerogel composite loaded with silica nanoparticles and calcium	Colloids Surfaces B Biointerfaces 2021;198:111454 Q1	In vitro and in vivo experimental study
Guangqian Lan et al., 2015 [30]	Chitosan/gelatin composite sponge is an absorbable surgical hemostatic agent	Colloids Surfaces B Biointerfaces 2015;136:1026–34. Q1	In vitro and in vivo experimental study
Padalhin AR, and Lee BT, 2019 [31]	Hemostasis and Bone Regeneration Using Chitosan/Gelatin-BCP Bi-layer Composite Material	ASAIO J. 2019;65(6):620–7. Q1	In vitro and in vivo experimental study

**Table 3 polymers-15-00575-t003:** Analysis of characterization methods, biocompatibility test and hemostatic test.

Studies	Materials Composition	Analysis Method	Conclusion
Jalal Ranjbar et al., 2021 [28]	Chitosan + gelatin + oxidized cellulose fibers (OF) by 2,2,6,6-tetramethylpiperidine-1-oxyl (TEMPO)	Characterizations Scanning electron microscope (SEM) Fourier transform infrared spectroscopy (FTIR) Swelling test Biocompatibility Cytotoxicity test Biodegradable test Hemostatic/coagulation test	The addition of TEMPO-oxidized cellulose fibers enhances the swelling ratio and clotting capabilities of CS/gelatin sponges. According to the MTT assays, the sponges were biocompatible. CS/gelatin/oxidized cellulose composite sponges have a greater bio-absorbability in rat liver. CS/gelatin composite sponges containing TEMPO-oxidized cellulose fibers are promising materials for use as hemostatic agents in low to moderate bleeding.
Gokul Patil et al., 2021 [29]	Chitosan + gelatin + silica nanoparticles + calcium	Characterizations Scanning electron microscope (SEM) Fourier transform infrared spectroscopy (FTIR) Swelling test Mechanical strength Biocompatibility Cytotoxicity test Shelf-life stability Hemostatic/coagulation test	The xerogel composite had a high absorption capacity and excellent clotting efficiency. In vivo test by application composite in lethal femoral artery injury in rats was very durable, withstands pressure, and quickly created a stable blood clot in 2.5 min. The composite could also be removed from the wound site without using saline or water. According to shelf-life experiments, the mixture preserved its beneficial characteristics for up to 1.5 years. The xerogel composite has the potential to save lives as a topical hemostatic agent.
Guangqian Lan et al., 2015 [30]	Chitosan + gelatin with various comparisons of material composition	4. Characterizations Scanning electron microscope (SEM) Fourier transform infrared spectroscopy (FTIR) Swelling test 5. Biocompatibility Cytotoxicity test Biodegradable test 6. Hemostatic test Coagulation test PT and aPTT Platelet adhesion Thrombin generation	Compared to chitosan and gelatin sponge alone, CG showed better hemostatic properties. It independently performs better in liquid absorption and platelet aggregation than in chitosan and gelatin sponges. Cytotoxicity assays revealed that CG produced increased cell proliferation with high cell viability and no evident cytotoxicity. CG has the potential to be employed in surgical hemostasis to minimize low-pressure bleeding.
Padalhin AR, and Lee BT, 2019 [31]	Chitosan + gelatin + bi-phasic calcium phosphate (BCP)	Characterizations Scanning electron microscope (SEM) Energy dispersive spectrometry (EDS) Biocompatibility Blood absorption test Cell adhesion and proliferation Hemostatic/coagulation test	A bi-layer composite topical hemostat made of chitosan hydrogel and the gelatin-BCP electrospun mat have been effectively manufactured and tested on bleeding bone. The bi-layer composite material was used to accomplish successful hemostasis by filtering cells on the electrospun layer and continuously absorbing blood serum within the hydrogel layer.

## Data Availability

Not applicable.

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
