# Peer review of "Characterization and Analysis of Chitosan-Gelatin Composite-Based Biomaterial Effectivity as Local Hemostatic Agent: A Systematic Review"

_polymers, 2023, doi:10.3390/polym15030575_

Round 1

Reviewer 1 Report

In this paper, the authors analyzed the characteristics of gelatin-chitosan composite-based biomaterial and its effectivity as a local hemostatic agent. The content is rich and substantial. However, there are still some issues to be addressed. The specific comments can be found as following:

1.     At the end of the introduction part, the author can add some general introductions about this work.

2.     Authors can add the serial numbers 2.1. and 2.2. before “Search Strategy and Paper Selection” and “Assessment of Methodological Quality and Risk of bias” in the Materials and Methods section, respectively, to make the paper structure clearer.

3.     The advantages of chitosan and gelatin should be further clarified with supporting recent articles: Recent advancements in applications of chitosan-based biomaterials for skin tissue engineering; Packaging and degradability properties of polyvinyl alcohol/gelatin nanocomposite films filled water hyacinth cellulose nanocrystals; etc.

4.     In Table 2, attention should be paid to the character spacing to make the format more standardized.

5.     In table 3, the conclusion can be provided with key points.

6.     In Figure 2, it should be noted that the format of initial letters is uniform.

7.     The format of references introduced in the text should be uniform. For example, the quotation format in line 95 is inconsistent with other quotation formats.

8.     It is a review article; however, authors wrote it in a style of research paper. Authors are suggested to rearrange the structure of this manuscript according to the style of review article.

9.     One more section of “Perspectives” or “Outlook” should be added to present the challenges and possible solutions to guide the future studies.

10. There are still some typos and grammar issues in the manuscript. Authors should carefully recheck the whole manuscript.

Author Response

Dear Reviewer 1,

Reviewer 2 Report

- This review was written based on four articles. Number of used articles too small.

- The objectives of the review is unclear.

- In practically, in this review duscussion is absent. It is very short and only includes a list of results.

- The manuscript cannot be published as it stands.

Author Response

Dear Reviewer 2,

Reviewer 3 Report

1. The author did not well express the necessity of this systematic review in the Introduction.

2. Aren't Scopus and Science Direct the same company's databases? Why not use WOS?

3. I don't think the author has made a reasonable analysis of these documents.

Author Response

Dear Reviewer 3,

Round 2

Reviewer 1 Report

Authors did not addressed this reivewers' comments well. There are only 26 references in this manuscript, which can not support a review article. From this point of view, the topic authors chose is not deserved to write a review article. In addition, the writing style of this manuscript is suitable for research article. 

Author Response

Dear Revier 1,

Reviewer 2 Report

accept in present form

Author Response

Dear Reviewer 2,

Reviewer 3 Report

The present form can be accepted.

Author Response

Dear Reviewer 3,

Round 3

Reviewer 1 Report

Although authors have added some new references, which still cannot support a review article. From this point of view, the topic authors chose is not deserved to write a review article. 

Author Response

Dear Reviewer 1,
